# Deep Signal-Dependent Denoising Noise Algorithm

**Lanfei Zhao** [1] , **Shijun Li** [1] **and Jun Wang** [2,*]

1   The Higher Educational Key Laboratory for Measuring & Control Technology and Instrumentations of Heilongjiang Province, Harbin University of Science and Technology, Harbin 150080, China
2   School of Information Engineering, Quzhou College of Technology, Quzhou 324000, China
*   Correspondence: 145688@qzct.edu.cn; Tel.: +86-181-0570-1939

**Abstract:** Although many existing noise parameter estimations of image signal-dependent noise have certain denoising effects, most methods are not ideal. There are some problems with these methods, such as poor noise suppression effects, smooth details, lack of flexible denoising ability, etc. To solve these problems, in this study, we propose a deep signal-dependent denoising noise algorithm. The algorithm combines the model method with a convolutional neural network. We use the noise level of the noise image and the noise image together as the input of the convolutional neural network to obtain a wider range of noise levels than the single noise image as the input. In the convolutional neural network, the deep features of the image are extracted by multi-layer residuals, which solves the difficult problem of training. Extensive experiments demonstrate that our noise parameter estimation has good denoising performance.

**Keywords:** signal-dependent noise; noise parameter estimation; convolutional neural network; image denoising

## 1. Introduction

Image denoising is a classic and indispensable research topic in low-level vision tasks, which is the premise of high-level vision tasks [1]. The clear and high-quality images obtained by denoising serve the high-level vision and make completing the tasks better. This paper mainly introduces the denoising of signal-dependent noisy images.

With the continuous updating of digital image acquisition technology, equipment for acquiring images and the number of acquired images are also increasing, and people's requirements for image quality are also increasing. However, due to the imperfections of imaging systems, imaging equipment, and transmission media, many different types of noise are introduced during the formation of images in different devices, which can affect the imaging effect, and thus the quality of the image. Noise-contaminated images will have a considerable impact on subsequent image processing. Noise pollution will affect the quality of subsequent image processing, such as image recognition, segmentation, classification, and so on [2].

In this paper, we model the signal-independent noise in the image, and then input it into the denoising network to solve the problem. During training and testing, networks that take the noise level of the input image and the noise image as input have a wider range of noise levels than networks with a single noise image as the input. The success of CNN denoisers is significantly dependent on whether the distributions of synthetic and real noises are well matched. Therefore, realistic noise models are the foremost issue for blind denoising of real photographs. According to [3], signal-dependent noise can be modeled by signal-dependent Gaussian noise distribution, and the signal-dependent Gaussian model has been considered as a more appropriate alternative than other models for real raw noise modeling. In addition, it is necessary to further optimize the graph after noise modeling. Because there are two unavoidable errors in modeling, the first one is the error generated in the estimation of noise parameters. The second is when pixel values are obtained by the

local statistical feature method, which also has errors. Thus, after acquiring the preliminary picture, we use the preliminary image and the noise image together as the input into the denoising network. The two errors are reduced through the network.

At present, image denoising relies on the image degradation model, but most of the image degradation model algorithms are simple and do not match the complex noise in the real image. It leads to the less ideal denoising effect. Therefore, it is necessary to design a more reasonable model or an algorithm with better performance. This paper proposes a method combining the model method and the convolutional neural network. The preliminary image obtained by the model method is used as a part of the network's input, and the final image is output through the continuous iteration of the network.

Most image denoising is Gaussian noise denoising. The variance of the noise is constant and does not change with the position and intensity of the pixel, the noise level is the only parameter required for modeling. The model of this degradation process is generally defined as:

$$y = x + n \tag{1}$$

where $x$ is the degraded image pixel value, $y$ is the original image pixel value, $n$ is Additive White Gaussian Noise (AWGN) with a standard deviation of $\sigma$. Since Gaussian white noise has only one unknown parameter, the variance of the noise, DnCNN enumerates the variance of different values in the process of training the network model [4], meaning that the network can remove AWGN in the case of unknown noise variance. However, because there are other noises in the real image, the Gaussian modeling of the image cannot achieve a good denoising effect.

Alessandro Foi [5] modeled the real noise as a signal-dependent Poisson–Gaussian noise model and proposed an estimation method for Poisson–Gaussian noise parameters. However, he trimmed the underexposure and overexposure data when experimenting. Despite achieving good performance, the method is limited to Poisson–Gaussian noise due to modeling problems. The model needs to know the Gaussian and Poisson components a and b, the model is as follows:

$$z(v) = k(v) + \sigma(k(v))\zeta(v) \tag{2}$$

$$\sigma^2(y(v)) = ay(v) + b \tag{3}$$

where $v$ is the pixel position in the domain. $z(v)$ is the observed signal, and $k(v)$ is the original signal. $\zeta(v)$ is zero-mean independent random noise with a standard deviation equal to 1. $\sigma$ is a function that gives the standard deviation of the overall noise component. The variance is an expression for the Poisson Gaussian noise component from which estimates of $a$ and $b$ are obtained.

Liu et al. [6] proposed a generalized signal-dependent noise model.

$$g = f + f^\gamma u + w \tag{4}$$

$$\sigma(f) = \sqrt{f^{2\gamma}\sigma_u^2 + \sigma_w^2} \tag{5}$$

where $g$ is the noisy pixel value, $f$ is the noise-free pixel value, and $u$ and $w$ are zero-mean Gaussian variables. $\gamma$ is an exponential parameter that controls signal dependence by changing the three NLF parameters $\gamma$, $\sigma_u^2$, and $\sigma_w^2$, and the noise model can represent various types of noise by changing these three parameters. Existing models usually assume that one parameter is known, and only estimate the remaining two parameters to simplify the problem. This method requires three parameters; thus, the amount of calculation is large.

The model we propose in this paper is a noise parameter estimation for real camera noise, which can be modeled with a signal-dependent Gaussian distribution [3].

$$x_p \sim N(y_p, \sigma_r^2 + \sigma_s y_p) \tag{6}$$

where $x_p$ is the degraded image pixel value, $y_p$ is the original image pixel value, and the noise parameters $\sigma_s$ and $\sigma_r$ are fixed and only change with the gain of the sensor. The main reasons for adopting this model are as follows: similar to the generalized noise model shown in [6], the exponential coefficient will make the calculation of the numerical solution difficult, and the number of unknown parameters in the Gaussian correlated noise model is less than that of the generalized signal-dependent noise model, thus the estimation difficulty will be much less. The model we use is more accurate in parameter estimation than the Poisson–Gaussian noise model, and the denoising effect is better.

In this paper, we propose a deep signal-dependent denoising noise algorithm. The algorithm further improves the noise suppression ability and improves the visual effect of the restored image. The denoising algorithm in this paper can effectively improve image quality, which plays an important role in many fields. First, we find a more reasonable model for the signal-related noise image, which has a wider range of application scenarios and a higher accuracy of parameter estimation. Second, we combine the model with the convolutional neural network to further improve the image-denoising effect. The algorithm further improves the noise suppression ability and improves the visual effect of the restored image. The denoising algorithm in this paper can effectively improve the image quality, which has a wide range of applications for the multimedia, military, medical, and other industries.

## 2. The Related Work

Ben et al. [3] proposed a technique for jointly denoising bursts of images taken from a handheld camera. Burst image noise parameter estimation operates on a set of successive, rapidly taken images to compute a single, noise-free result. In particular, they propose a convolutional neural network architecture for predicting spatially varying kernels that can both align and denoise frames: a synthetic data generation approach based on a realistic noise formation model. The method for burst denoising they proposed has the signal-to-noise ratio benefits of multi-image denoising and the large capacity and generality of a convolution neural network.

Liu et al. [7] proposed a segmentation-based image denoising algorithm for signal dependent noise. First, they identify the noise level function for a given single noisy image. Then, after initial denoising, segmentation is applied to the prefiltered image.

Tan et al. [8] proposed a deep convolutional neural network named "deep residual noise estimator" (DRNE) for pixel-wise noise-level estimation. The DRNE framework they designed consists of a stack of customized residual blocks without any pooling or interpolation operations. The noise level estimation graph obtained by DRNE, and the ground truth graph are used as the input of the convolutional neural network, which makes the original network achieve better results.

Talmaj et al. [9] improved KPN, and they proposed a deep neural network-based approach called Multi-Kernel Prediction Networks (MKPN) for burst image denoising. MKPN predicts kernels of not just one size but of varying sizes and performs a fusion of these different kernels resulting in one kernel per pixel. This enables MKPN to better extract information from images. MKPN has achieved good results in texture and homogeneous area denoising. Bingyang et al. [10] proposed a novel network including noise estimation module and removal module (NERNet). The noise estimation module automatically estimates the noise level map corresponding to the information extracted by symmetric dilated block and pyramid feature fusion block. To fuse noisy multi-modality image pairs accurately and efficiently, Huanqiu et al. [11] proposed a multi-modality image simultaneous fusion and denoising method.

Zhonghua et al. [12] proposed a novel model-guided boosting framework. By using the Regularization by Denoising (RED), they could apply explicit regularization equipped with powerful image denoising engine to establish the global minimization problem, making the obtained model clearly defined and well optimized. The framework enjoyed

the advantage of easily extending to the case of composite denoising via superadding a regularization term.

Gang Liu et al. [13] proposed a true wide CNN (WCNN) to reorganize several convolutional layers. Each subnetwork had its own input and output and was supervised by its own loss function to capture image features with a specific direction and scale, allowing the WCNN to have sufficient convolutional layers to capture image features while avoiding vanishing/exploding gradients. Jiechao Shen et al. [14] proposed the sparse representation-based network (SRNet). They considered combining the sparse representation with deep learning to make this traditional model more effective and efficient. They embedded the convolutional neural network (CNN) into the sparse representation framework. Laya et al. [15] proposed the Multi Scaling Aided Double Decker (MUS-ADD) convolutional neural network. It solved the disadvantages of the traditional method, which required a large number of models, etc. Shuang Xu et al. [16] built a more interpretable network. An observation model was proposed to account for modality gap between target and guidance images. Then, they formulated a deep prior regularized optimization problem, and solved it by the alternating direction method of multipliers (ADMM) algorithm. Yang Ou et al. [17] proposed a novel multi-scale weighted group sparse coding model (MS-WGSC). It better restored the structure and the edges of images contaminated by noise. Lei Zhang et al. [18] proposed the Robust Low-Rank Analysis with Adaptive Weighted Tensor (AWTD) method. It obtained the low-rank approximation of the tensor by adding adaptive weights to the unfolding matrix of the tensor. By decomposing true image into a cartoon part and texture part, Xiao Li et al. [19] proposed a fractional image denoising model with two-component regularization terms. Sunder Ali et al. [20] proposed a cascaded and recursive convolutional neural network (CRCNN) framework, which could cope with spatial variant noise and blur artifacts in a single denoising framework. Lei Zhang et al. [21] designed a novel denoising model named Kronecker Component with Low-Rank Dictionary (KCLD), which replaced the Frobenius norm with a Nuclear norm in order to capture the low-rank property better. Phan et al. [22] proposed an adaptive model that used the mean curvature of an image surface to control the strength of smoothing. A fast method for noise level estimation is proposed to improve the effectiveness of the proposed model.

In [23], Heng Yao et al. proposed an algorithm to efficiently estimate the noise level function (NLF), which is defined as the noise standard deviation with respect to image intensity. The method divided the input image into overlapping patches. The confidence levels of the noise samples and the prior of the camera response function were then employed as constraints for the recovery of the NLF. The NLF was incorporated into other denoising schemes to obtain better results than the original scheme.

Guo et al. [24] trained a convolutional blind denoising network (CBDNet) with a more realistic noise model and real-world noisy–clean image pairs. To further provide an interactive strategy to rectify denoising results conveniently, a noise estimation subnetwork with asymmetric learning to suppress under-estimation of noise level was embedded into CBDNet. They adopted an asymmetric loss by imposing more penalty on an under-estimation error of noise level, making CBDNet perform robustly when the noise model was not well matched with real-world noise.

Zhao et al. [25] proposed a simple but robust network called SDNet to improve the effectiveness and practicability of deep denoising models. Additionally, there is no need to estimate the noise level. The model learning had exploited clean–noisy image pairs, newly produced, built on a generalized signal-dependent noise model. To separate the noise from image content as fully as possible, a kind of lifting residual module was specifically proposed for discriminative feature extraction. The network emphasized separating the noise from image content via a direct, fully end-to-end residual learning strategy.

In [26], Dong et al. proposed a grouped residual dense network (GRDN). The core part of DRDN is defined as a grouped residual dense block (GRDB) and used as a building module of GRDN. They experimentally show that image-denoising performance can be

significantly improved by cascading GRDBs. Meanwhile, they developed a new generative adversarial network-based real-world noise modeling method.

## 3. Proposed Methods

Synthetic noise images are used in this paper. The most important noises in real images are Poisson noise (shot noise) and Gaussian noise. Some scholars have proposed a simple and practical noise model based on the original data of digital imaging sensors [3]. The signal-dependent noise model gives a function consisting of the Poisson part of the modeled photon sensing and the Gaussian part of the remaining stationary perturbations in the output data. Two parameters of a given noisy image model can be estimated by the image restoration algorithm based on local statistical features. Overall architecture of the algorithm is shown in Figure 1.

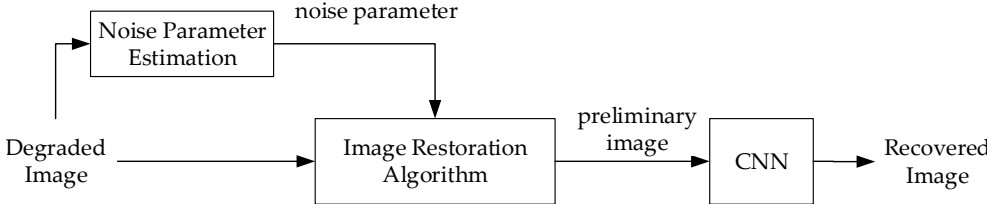

**Figure 1.** Overall architecture of the algorithm.

### 3.1. The Image Restoration Algorithm on Local Statistical Features

In this paper, the Gaussian correlation noise model shown in Equation (6) is used to model the image degradation process. The estimated values of the two parameters are obtained by estimating the parameters of the noise model, and then the preliminary image is obtained through local statistical features.

### 3.1.1. Noise Model Parameter Estimation

First, two important assumptions are given: first, according to [27], the statistical features of a pixel in a degraded image with signal-dependent noise can be derived from a local region centered at that pixel. Second, there are a certain number of locally homomorphic sub-blocks in the original image, and the variance of the pixel intensity in the sub-blocks is approximately 0. Assuming that the pixel blocks of $y_p$ and its adjacent areas are homomorphic sub-blocks, the brightness of the original pixels in the homomorphic sub-blocks are $y_p$ and $E(y_p) = y_p$. The equation represents the local mean of the homomorphic subblock image without noise as equal to the local mean of noisy image. According to the modeling formula of $x_p$, we can acquire:

$$D(x_p) = \sigma_r^2 + {\sigma_s}^2 y_p \tag{7}$$

$$E(x_p) = y_p \tag{8}$$

where $E$ is local mean, $D$ is variance of noise, $x_p$ is the degraded image pixel value, $y_p$ is the original image pixel value, and $\sigma_s$ and $\sigma_r$ are the noise parameters. The two formulas can be simplified to acquire:

$$D(x_p) = \sigma_r^2 + {\sigma_s}^2 E(x_p) \tag{9}$$

Equation (9) is derived from Equations (7) and (8). Equation (7) is a quadratic equation of $\sigma_s$ and $\sigma_r$, which has an infinite number of solutions about the variables $\sigma_s$ and $\sigma_r$. To obtain the unique solution of $\sigma_s$ and $\sigma_r$, it is necessary to select multiple homomorphic sub-blocks, then a set of linear equations composed of binary linear equations in the form of Equation (9) can be obtained. The number of sub-equations depends on the homomorphic sub-blocks quantity. Typically, the number of image homomorphic sub-blocks are more

than two, thus the system of equations is an overdetermined system of equations. The overdetermined systems of equations are expressed.

$$\sum_{p \in H} \left[ \sigma_r^2 + {\sigma_s}^2 E(x_p) - D(x_p) \right]^2 = 0 \tag{10}$$

In this paper, Equation (10) is solved by the least squares method. By taking the partial derivative of $\sigma_s$ and $\sigma_r$ in Equation (10), we can acquire both of the equations in Equation (11). The solutions of $\sigma_s$ and $\sigma_r$ are transformed into solving the system of linear equations shown below. We polluted the images of the Set12 dataset with noise $\sigma_s = 2.0 \times 10^{-2}$, $\sigma_r = 10^{-1}$, and then estimated the average value of noise parameters obtained by Equation (11) to be $\sigma_s = 1.978 \times 10^{-2}$, $\sigma_r = 1.15 \times 10^{-1}$, with an error within 0.02.

$$\begin{cases} \sum_{p \in H} \left[ \sigma_r^3 + {\sigma_s}^2 \sigma_r E(x_p) - \sigma_r D(x_p) \right] = 0 \\ \sum_{p \in H} \left[ \sigma_s \sigma_r^2 E(x_p) + \sigma_s^3 E^2(x_p) - \sigma_s D(x_p) E(x_p) \right] = 0 \end{cases} \tag{11}$$

At this point, the unknown parameters $\sigma_s$ and $\sigma_r$ of the model have unique solutions that satisfy the least squares method.

### 3.1.2. The Image Restoration Algorithm on Local Statistical Features

It can be seen from the above reasoning that the variance corresponding to the observations of the Gaussian signal-dependent noise model is as follows:

$$\sigma^2(x_p) = \sigma_r^2 + \sigma_s^2 y_p \tag{12}$$

Arranging the above formula will result in $y_p = \frac{\sigma^2(x_p) - \sigma_r^2}{\sigma_s^2}$. Since the parameters $\sigma_s$ and $\sigma_r$ can be estimated by solving the above equations, just solve for $\sigma^2(x_p)$ to acquire the preliminary image. According to the assumption proposed by Lee [28], the mean $\mu(x_p)$ and variance $\sigma^2(x_p)$ of $x_p$ are determined by the mean and variance of the observations in the neighborhood pixels point of the center pixel $p$. Thus, $\sigma^2(x_p)$ can be obtained as follows:

$$\sigma^2(x_p) = \frac{1}{M_t} \sum_{q \in W_p} (x_q - \mu(x_p))^2 \tag{13}$$

where $M_t$ is the number of pixels in the neighborhood. Substituting the expression of $\sigma^2(y_p)$ into the above formula will result in:

$$y_p = \frac{1}{\sigma_s^2 M_t} \sum_{q \in W_p} (x_q - \mu(x_p))^2 - \frac{\sigma_r^2}{\sigma_s^2} \tag{14}$$

### 3.2. Convolutional Neural Network

The preliminary image obtained by the above image restoration algorithm based on the statistical characteristics of the local regions inevitably has two errors. The first error is caused when estimating the parameters $\sigma_s$ and $\sigma_r$. The second error is brought about by the local statistical features of the selected pixels. Therefore, the denoising effects of the preliminary image are not ideal. The following two errors caused by the above algorithm are reduced by the convolutional neural network.

#### 3.2.1. Network Design

The network in this paper employed a simple design, which can reduce the amount of calculation and speed up the training while completing the task. The function of the designed network is mainly to reduce the error of the preliminary image. The convolutional network structure is illustrated in Figure 2. The network takes the preliminary image and noisy image as input, and then extracts the shallow features of the image by a convolutional layer, and then extracts the deep features of the image through the residual network. The residual network is composed of a set of 6 residual blocks. Deeper networks are used

to extract deep features. However, after the network layer is deepened, the gradient disappearance and gradient explosion make the training of the network meaningless. Therefore, the residual network is used to avoid the problem, which not only ensures the effect but also controls speed. After the residual network extracts the deep-level features, the data are up_sampled using a up_convolutional layer and the image is returned to the original size. The last convolution layer fuses the features into a single-channel image. The size of the convolution kernel used in the first convolutional layer and the last convolutional layer are both 9 × 9, and the size of the convolution kernel used by all the remaining convolutional layers is 3 × 3. The convolution layers in the residuals block are all 128 convolution kernels with a stride of 1. Only in this way can each layer in the residual network output the same size as the feature map. It helps to fuse features. The LeakyReLU activation function with a slope of 0.2 is employed in the paper, which fixes the problem that the non-positive gradient of ReLU is 0, causing some parameters never to be updated. This activation function has a small positive slope in the negative area, thus it can backpropagate even with negative input values. Furthermore, after experiments, it is found that batch normalization is hardly helpful for the denoising of real images. Part of the reason is that the real noise distribution is fundamentally different from the Gaussian distribution. The architecture of the proposed network is shown in Figure 3a, and the residual connection is shown in Figure 3b. Table 1 shows the parameter settings of Convolutional Neural Networks.

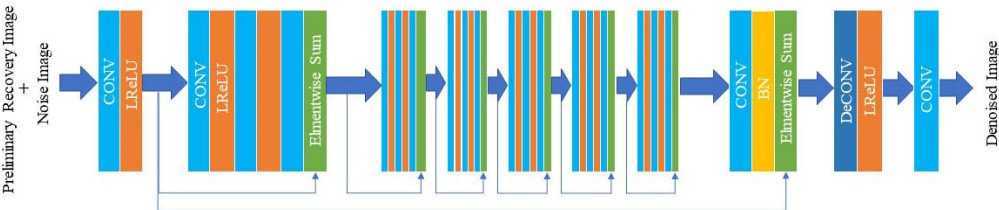

**Figure 2.** Convolutional network diagram.

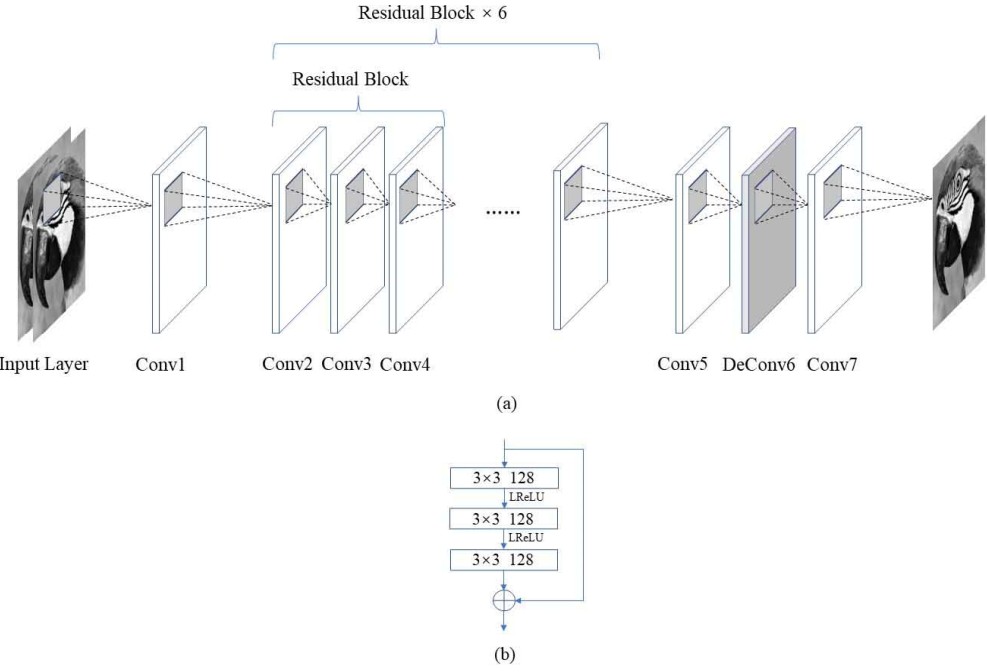

**Figure 3.** The designed network details of each convolutional layer. (**a**) Convolutional neural network structure. (**b**) Residual connection.

**Table 1.** The parameter settings of Convolutional Neural Network.

| Layer | Kernel | Leaky-ReLU | Stride |
|---|---|---|---|
| Conv1 | $9 \times 9 \times 64$ | Yes | 1 |
| Conv2 | $3 \times 3 \times 128$ | Yes | 1 |
| Conv3 | $3 \times 3 \times 128$ | Yes | 1 |
| Conv4 | $3 \times 3 \times 128$ | No | 1 |
| Conv5 | $3 \times 3 \times 128$ | No | 1 |
| DeConv6 | $3 \times 3 \times 256$ | Yes | 1 |
| Conv7 | $9 \times 9 \times 1$ | No | 1 |

### 3.2.2. Loss Function Design

The mean squared error (MSE) below is used as the objective function (the loss function) to measure the difference between the predicted image $R(Y_j)$ and the corresponding ground truth value $Y_j - X_j$. $X_j$ represents the $j$th clean image. $\theta$ represents the parameters of the trained model in the network. $\left\{ (Y_j, X_j) \right\}_{j=1}^{N}$ represents there are N image pairs of predicted images and real images. Potentially clean images are recovered by the Adam optimizer using a loss function.

$$L(\theta) = \frac{1}{2N} \sum_{j=1}^{N} \left\| R(Y_j, \theta) - (Y_j - X_j) \right\|_2^2 \tag{15}$$

### 4. Experiment

In this section, we assess the performance of the proposed method and compare it with other methods to show the improvement of the method relative to other methods.

The experimental environment comprised an Intel Core i7-9700k CPU, 128 GB RAM, and a NVIDIA GeForce RTX 2070 GPU. In addition, the deep learning framework used was Pytorch.

The paper uses the Berkeley Segmentation Dataset (Berkeley segmentation dataset, BSD) (BSD500 for short) as the network training dataset. The BSD500 dataset contains 500 grayscale images, and all images are of size $481 \times 321$. In the experiment, 200 sets of noise are added to each image to construct an image with real noise. Therefore, the training dataset consists of $10^5$ training samples in total. The noise parameters $\sigma_s$ and $\sigma_r$ are randomly sampled from $[10^{-3}, 10^{-2}]$ to $[10^{-2.5}, 10^{-1}]$, respectively.

In the selection of test data sets, we use Kodak, McMaster, and Set12 as test data sets, and compare the more advanced methods proposed on these data sets. Although KPN [3] is designed for multi-frame image input, it can be adjusted to a single-frame image by changing the network input to compare with our method.

For comparison with other methods, we evaluated the noise levels corresponding to three fixed sets of $\sigma_s$ and $\sigma_r$. Similar to [29], gamma correction was performed first, then we added signal-dependent Gaussian noise, took the generated noise image as the input: $N(0, \sigma_r^2 + \sigma_s y_p)$, where $y_p$ was the intensity of image pixels and the noise parameters $\sigma_s$ and $\sigma_r$ were randomly sampled from $[10^{-3}, 10^{-2}]$ to $[10^{-2.5}, 10^{-1}]$. Furthermore, similar to [4], the noise level was estimated as: $\sqrt{\sigma_r^2 + \sigma_s y_p}$. Note that the noise Equation (6) assigned three single sets of parameter values, i.e., $\sigma_1$ ($\sigma_s = 2.0 \times 10^{-2}$, $\sigma_r = 10^{-1}$), $\sigma_2$ ($\sigma_s = 6.0 \times 10^{-2}$, $\sigma_r = 2 \times 10^{-1}$), and $\sigma_3$ ($\sigma_s = 9.0 \times 10^{-2}$, $\sigma_r = 3 \times 10^{-1}$), to generate clean-noisy image pairs for image denoising. We claim that harnessing noise Equation (6) with combinations of different parameter settings may lead to better results. In spite of that, it was found that the parameter setting as above is a more robust candidate than several other choices.

Similar to most CNNs used for image denoising, the optimizer chooses the adaptive moment estimation (Adam) method in setting network parameters, and the loss function is

MSE. The whole training process contains 180 epochs, and the batch size is set to 20, thus the training data set can be divided into 5000 batches. However, for each epoch, we only sample the first 3000 batches for training. Before the next epoch training, we shuffle the order of all batches and select the top 3000 batches for training. The training adopts a linear piecewise learning rate, the learning rate of the top 90 epochs is $10^{-3}$, the next 60 epochs are $10^{-4}$, and the last 30 epochs are $10^{-5}$.

To test the effectiveness of each block for our network on image denoising, we used ablation experiments to analyze the performance. The images in the Set12 dataset were selected for the experiment, and the noise level $\sigma_1$ ($\sigma_s = 2.0 \times 10^{-2}$, $\sigma_r = 10^{-1}$) was added. Then, we denoised the images separately in groups. The denoising results are shown in Table 2, which shows the PSNR. The results show that residuals made the denoising effect better. Additionally, the addition of preliminary image made the denoising effect better.

**Table 2.** The average PSNR(DB) results of different methods on three datasets. (a) Remove three residual blocks. (b) Remove all residual blocks. (c) Remove the preliminary image from the input. (d) Original network.

| Method | a | b | c | d |
|:------:|:---:|:---:|:---:|:---:|
| PSNR | 28.67 | 29.61 | 31.24 | 33.50 |

The paper selects four methods: KPN [3], CBDNet [24], SRNet [14], and WCNN [13]. The link to the code is posted in references [30–33]. Five methods are compared on all the datasets used in the paper. KPN generates a stack of per-pixel filter kernels that jointly aligns, averages, and denoises a burst to produce a clean version of the reference frame. The establishment of the CBD model takes into account both the Poisson–Gaussian model and the in-camera processing pipeline. However, the Poisson–Gaussian model does not have a wider range of scenarios than the signal-dependent Gaussian model in the paper. Each subnetwork of WCNN has its input and output and is supervised by its loss function to capture image features with a specific direction and scale, allowing the WCNN to have sufficient convolutional layers to capture image features while avoiding vanishing/exploding gradients. SRNet embeds the CNN into the sparse representation framework. In each phase, two subnetworks are designed with MSR block to model the updating of the sparse coefficient and image, respectively. However, it suffers from performance bottlenecks and large time consumption.

Figure 4a–d is a diagram of each stage of the paper. To see the contrast, we enlarge the green part of the image into the red part. It can be seen that the images have more detail loss and less clarity even if the preliminary image can see the denoising effect. The denoising effect of the restored image is better. The picture detail loss is less, the recovery effect is better.

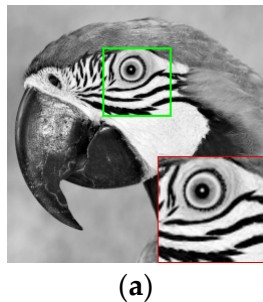 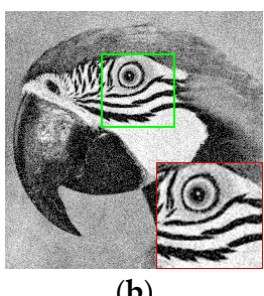 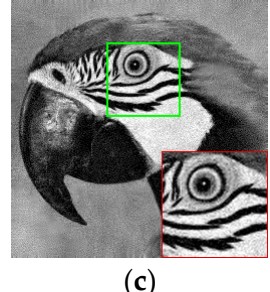 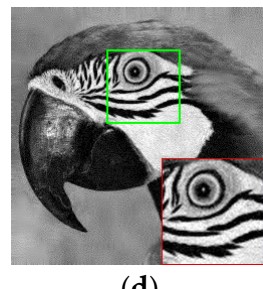

(**a**)  (**b**)  (**c**)  (**d**)

**Figure 4.** The denoising process of our algorithm. (**a**) Original image. (**b**) Noise image ($\sigma_3$ ($\sigma_s = 9.0 \times 10^{-2}$, $\sigma_r = 3 \times 10^{-1}$)). (**c**) The preliminary image. (**d**) The final image.

Figure 5a–e shows a visual comparison of different algorithms in the Set12 dataset. The KPN image effect is normal, the details are lost, and the denoising effect of the shadow area is not very ideal. CBDNet is better than KPN in detail recovery. It loses less detail

and restores the shadow part well, but the overall image is not clear. CBDNet is better at processing low-noise images. While SRNet and WCNN performed well in terms of image sharpness and detail, there is still room for improvement. The algorithm proposed in this paper can recover image details more clearly and obtain a better visual effect. It still does not effectively remove all real noise, because the noise model does not perfectly represent real noise.

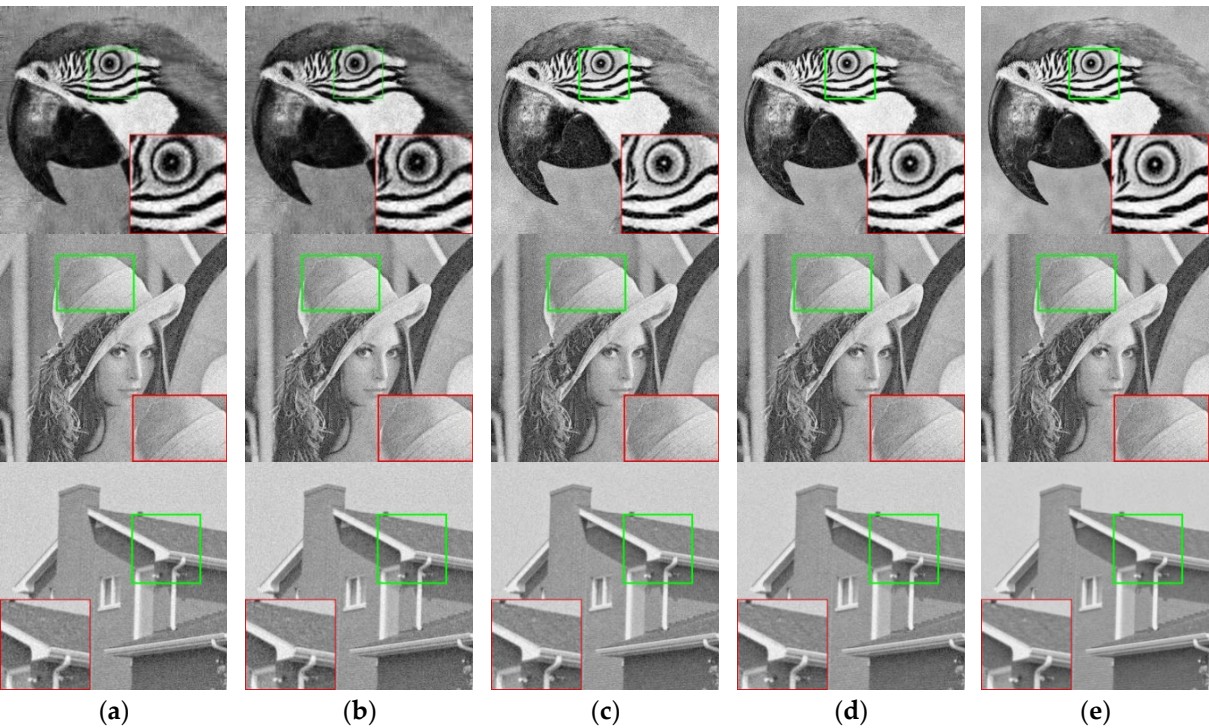

**Figure 5.** Denoising results of the images on the Set12 dataset by different methods. (**a**) KPN [3]. (**b**) CBDNet [24]. (**c**) SRNet [14]. (**d**) WCNN [13]. (**e**) Ours.

Table 3 shows the average PSNR in Set12 data sets. Table 4 shows the average SSIM of different algorithms in Set12 data sets. The average PSNR of the algorithm in the paper is higher than the other algorithms. Except for low-noise images, the denoising effect of CBDNet is slightly better than that of the method in this paper. The performance of SRNet and WCNN is similar to that of the proposed algorithm. However, in terms of SSIM, our algorithm is superior to other algorithms. It shows that the proposed algorithm is superior to the previous algorithms in denoising performance. The average SSIM index of the restored image is close to that of the original image, indicating that the quality of the restored image is close to that of the original noiseless image.

**Table 3.** The average PSNR(DB) results of different methods on three datasets.

| Dataset | Noise Level | KPN [3] | CBDNet [24] | SRNet [14] | WCNN [13] | Ours |
|---------|-------------|---------|-------------|------------|-----------|------|
| McMaster | σ1 | 32.69 | 34.34 | 33.76 | 33.65 | 34.31 |
|          | σ2 | 31.09 | 31.51 | 31.73 | 31.92 | 33.25 |
|          | σ3 | 29.76 | 29.84 | 30.42 | 30.78 | 31.03 |
| Set12   | σ1 | 30.96 | 33.53 | 33.11 | 32.91 | 33.50 |
|         | σ2 | 29.82 | 30.88 | 31.16 | 31.23 | 31.52 |
|         | σ3 | 28.59 | 28.79 | 29.28 | 29.57 | 29.95 |
| Kodak   | σ1 | 31.98 | 35.71 | 34.95 | 34.71 | 35.65 |
|         | σ2 | 30.57 | 31.86 | 32.44 | 32.93 | 33.63 |
|         | σ3 | 29.45 | 29.79 | 30.07 | 30.48 | 31.46 |

**Table 4.** The average SSIM results of different methods on three datasets.

| Dataset | Noise Level | KPN [3] | CBDNet [24] | SRNet [14] | WCNN [13] | Ours |
|---------|-------------|---------|-------------|------------|-----------|------|
| McMaster | $\sigma 1$ | 0.684 | 0.785 | 0.727 | 0.723 | 0.783 |
| | $\sigma 2$ | 0.615 | 0.661 | 0.689 | 0.697 | 0.715 |
| | $\sigma 3$ | 0.564 | 0.613 | 0.621 | 0.635 | 0.651 |
| Set12 | $\sigma 1$ | 0.668 | 0.729 | 0.710 | 0.701 | 0.724 |
| | $\sigma 2$ | 0.632 | 0.655 | 0.642 | 0.646 | 0.708 |
| | $\sigma 3$ | 0.551 | 0.577 | 0.598 | 0.585 | 0.604 |
| Kodak | $\sigma 1$ | 0.718 | 0.789 | 0.771 | 0.762 | 0.783 |
| | $\sigma 2$ | 0.684 | 0.724 | 0.735 | 0.738 | 0.761 |
| | $\sigma 3$ | 0.642 | 0.665 | 0.689 | 0.691 | 0.710 |

Figure 6 shows the average PSNR of our algorithm on the Set12 test set at the noise level of $\sigma_3$ ($\sigma_s = 9.0 \times 10^{-2}$, $\sigma_r = 3 \times 10^{-1}$). It shows the line graph of PSNR changing with the number of iterations. As the number of iterations increases, PSNR gradually flattens out.

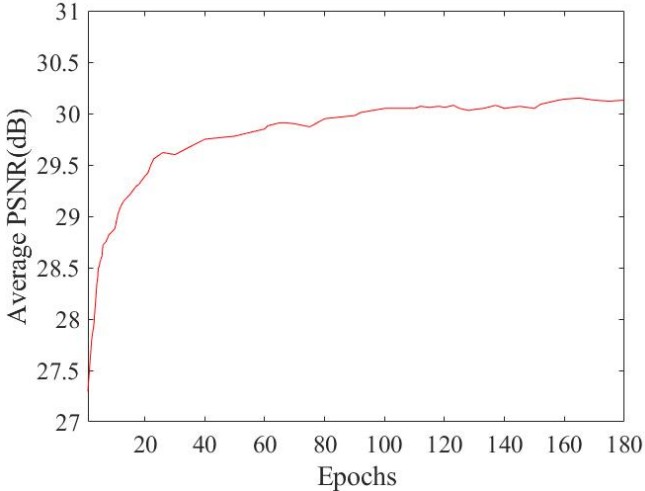

**Figure 6.** The relationship between the number of iterations Epoch and PSNR.

Figure 7 shows the loss of our algorithm on the Set12 test set at the noise level of $\sigma_3$ ($\sigma_s = 9.0 \times 10^{-2}$, $\sigma_r = 3 \times 10^{-1}$). It shows the line graph of loss changing with the number of iterations. As the number of iterations increases, loss decreases and gradually flattens out.

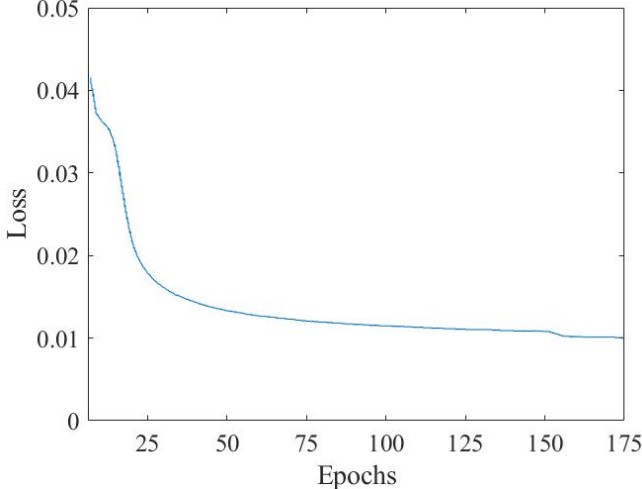

**Figure 7.** The relationship between the number of iterations of Epoch and Loss.

To verify the denoising effect of our algorithm under high noise, we added a set of denoising experiments $\sigma$ ($\sigma_s = 9.0 \times 10^{-1}$, $\sigma_r = 3.0 \times 1$) for verification. Figure 8a,b shows a comparison of the denoising effect of the proposed method in the case of high noise.

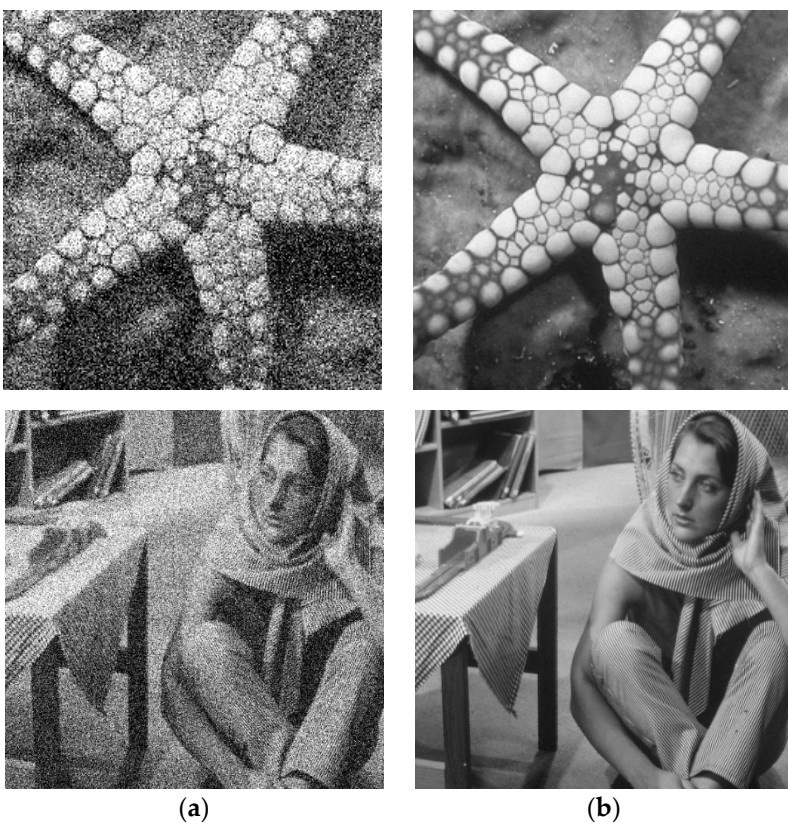

(**a**)　　　　　　　　　　　　　　　　　(**b**)

**Figure 8.** Image denoising comparison under high noise $\sigma$ ($\sigma_s = 9.0 \times 10^{-1}$, $\sigma_r = 3.0 \times 1$). (**a**) High noise image. (**b**) Denoising image.

To verify the performance of the denoising algorithm in real scenes, we selected the $\alpha$57 digital cameras launched by Sony to collect 30 images. To evaluate the performance of our method in terms of visual quality, we performed a subjective evaluation as a qualitative analysis experiment. The evaluation was carried out as a rating experiment where 40 users experienced in image processing viewed, in random order, a set of images. The users were asked to provide ratings for each image according to the following attributes: noise and detail. The final result of the ratings averaged over the observers is illustrated in Figures 9 and 10.

Figure 11a–f shows two images we randomly selected. From the first picture, the details of the houses in the shaded part of the image restored by KPN are missing, and the overall effect of the image is not ideal. The local detail of the CBDNet algorithm is stronger than KPN, but the overall visual effect is a bit fuzzy, and the image quality is not good. WCNN and SRNet are better than the first two algorithms in terms of sharpness and image detail, but there is still room for improvement. In the marked areas in the image, we can see that other algorithms are rough in detail recovery in the shadow area, and the denoising effect is not ideal. Our algorithm performs well in this respect. The algorithm in this paper has the strongest local detail retention ability, and the restored image has fine details and a clear texture. In the second picture, KPN is missing some details in the recovered image. The text in the book is obscured, and the overall picture is poor in definition. In the image recovered by CBDNet, the text of the book can be seen, but it is not clear, and the overall image is a little fuzzy. The images recovered by WCNN and SRNet are good, but lower than the algorithm in this paper. In the marked areas in the images, we can see that the labels in the images recovered by other algorithms are fuzzy, and the denoising effect around the

labels is not ideal, while the label recovered by our algorithm is clear and the denoising effect is ideal. The algorithm in this paper has a good performance in detail recovery and overall quality.

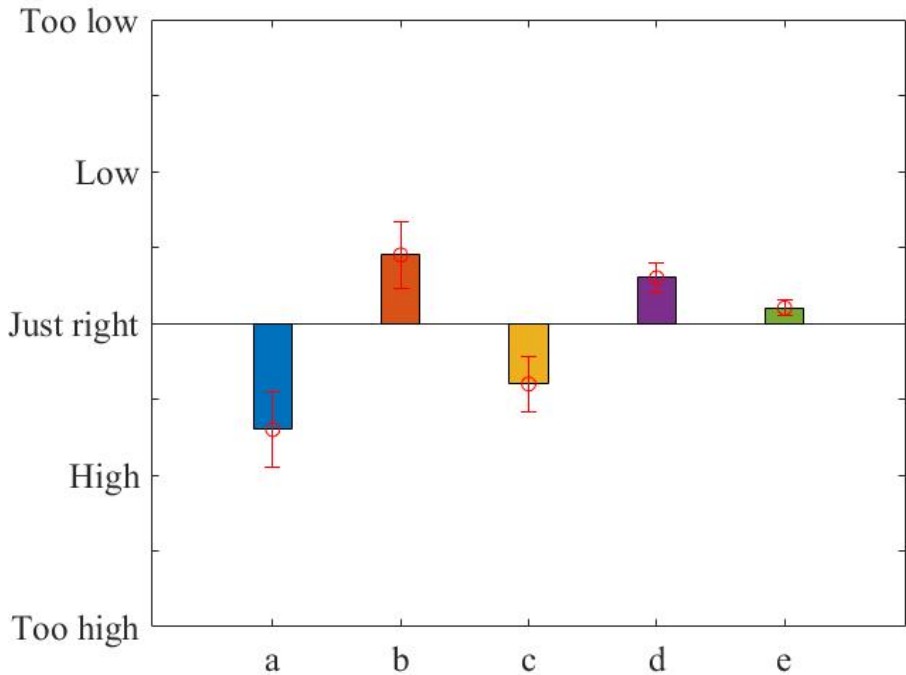

**Figure 9.** Qualitative analysis result of noise, showing the average ratings from the conducted experiment, with error bars for the standard errors. (**a**) KPN [3]. (**b**) CBDNet [24]. (**c**) SRNet [14]. (**d**) WCNN [13]. (**e**) Ours.

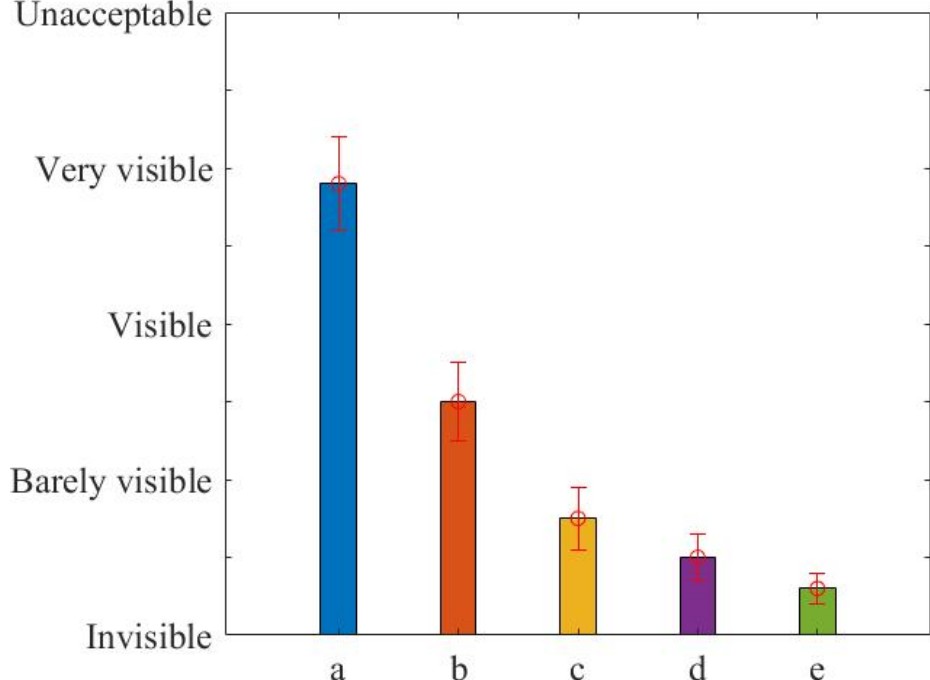

**Figure 10.** Qualitative analysis result of detail, showing the average ratings from the conducted experiment, with error bars for the standard errors. (**a**) KPN [3]. (**b**) CBDNet [24]. (**c**) SRNet [14]. (**d**) WCNN [13]. (**e**) Ours.

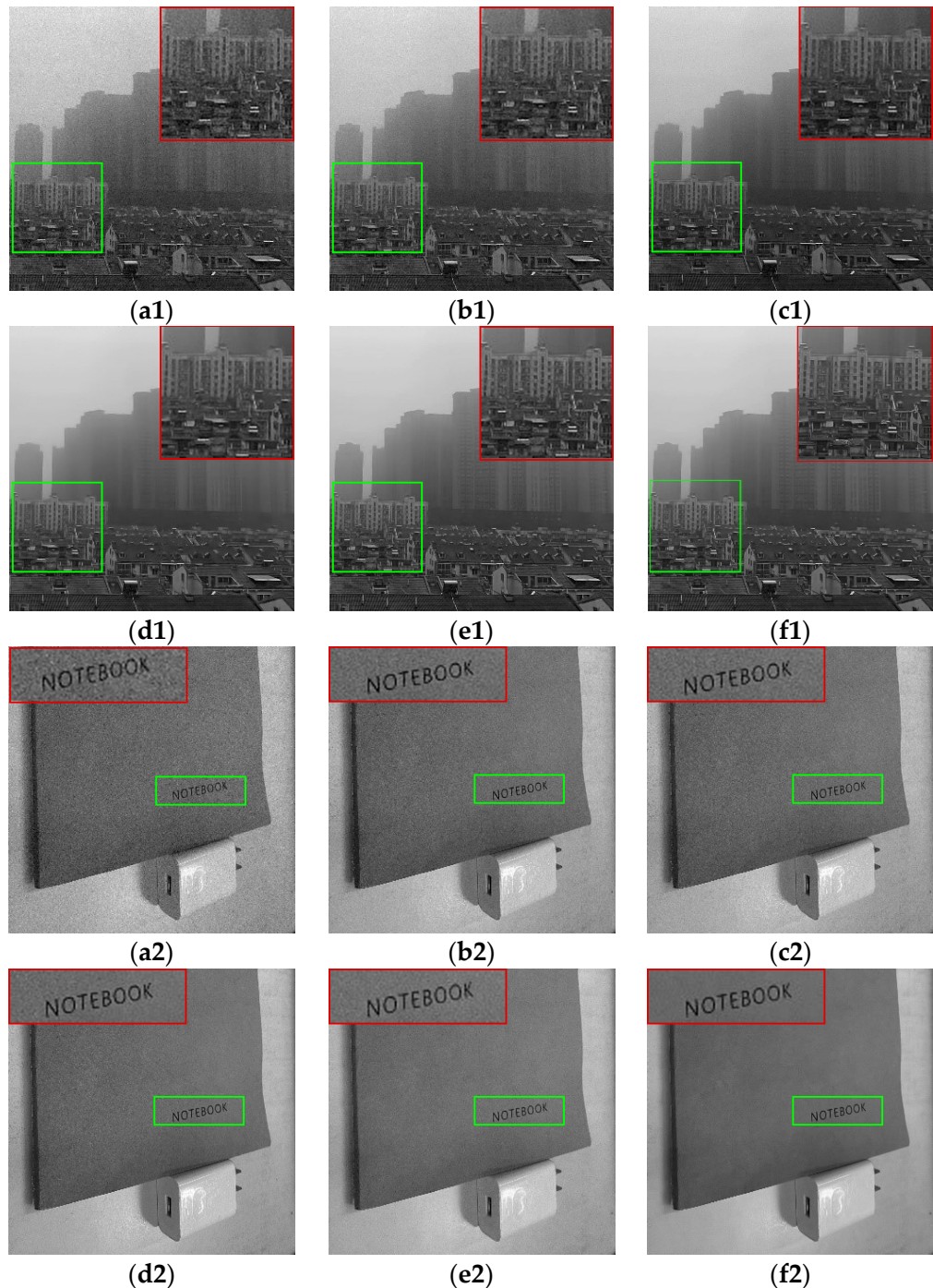

**Figure 11.** Image denoising comparison in real scenes. (**a1,a2**) The original image. (**b1,b2**) KPN [3]. (**c1,c2**) CBDNet [24]. (**d1,d2**) SRNet [14]. (**e1,e2**) WCNN [13]. (**f1,f2**) Ours.

Figure 12a,b shows the comparison before and after denoising of our algorithm. It can be intuitively seen from comparison figures that the noise in the images is greatly reduced after denoising by our algorithm. The image recovery effect is good, and the details are perfectly restored.

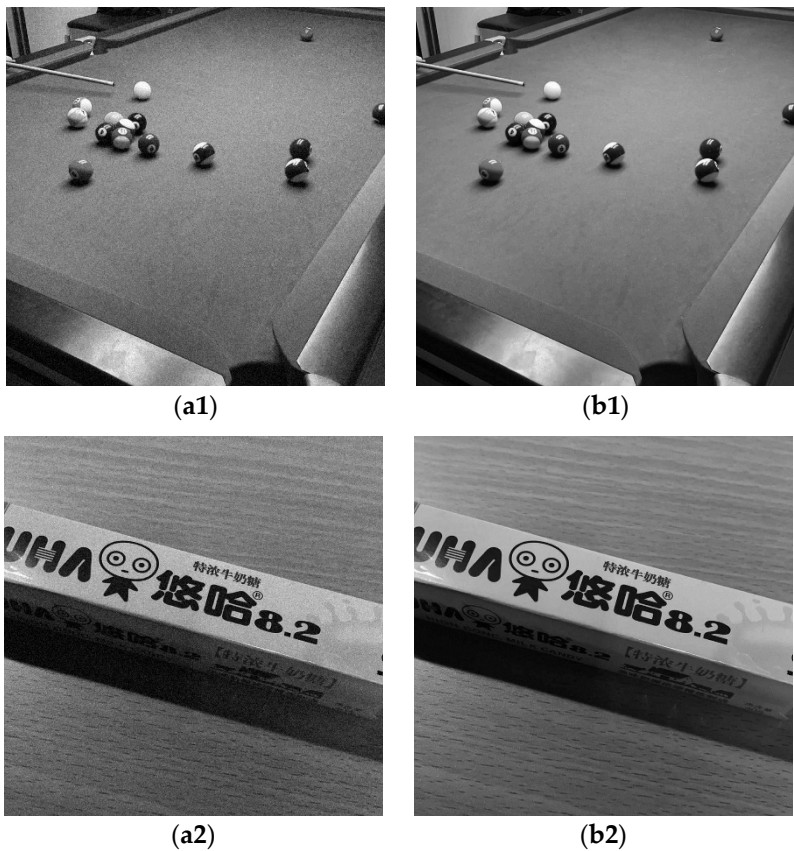

**Figure 12.** Image denoising comparison. (**a1**,**a2**) The original image. (**b1**,**b2**) Ours.

In general, our algorithm can reconstruct the detailed features more successfully. Although it does not achieve the best denoising effect in low noise images, the overall denoising effect is better and the restored images are clear, which is better than other algorithms.

## 5. Conclusions

The paper presents a deep signal-dependent denoising noise algorithm. There are two main stages to the work. Firstly, the noise parameters of the image are estimated by the noise parameters, and then the preliminary image is obtained according to the local statistical characteristics. Secondly, using the initial recovery map as input to the network along with the original map, take the initial restored image and the original image together as the input of the network, and the convolutional neural network can be used to reduce the error of the preliminary image and make the recovered image closer to the original image. We use the noise level of the noise image and the noise image together as the input of the convolutional neural network to obtain a wider range of noise levels than the single noise image as the input. It makes the denoising algorithm more widely used and has a better effect on detail recovery. We compared and analyzed denoising algorithms based on subjective vision. It was verified that the algorithm in this paper has a relatively evident inhibitory effect on signal-dependent noise, and the recovered images are of higher quality. This paper deals with grayscale images. For the future work, we hope to denoise color noise images. We believe that a combination of traditional image denoising and deep learning will become a great way to deal with noisy images.

**Author Contributions:** Conceptualization, L.Z.; methodology, L.Z. and J.W.; software, S.L.; validation, L.Z., S.L. and J.W.; formal analysis, S.L.; investigation, S.L.; resources, J.W.; data curation, L.Z.; writing—original draft preparation, S.L.; writing—review and editing, J.W.; visualization, J.W.; supervision, L.Z.; project administration, L.Z. and J.W.; funding acquisition, L.Z. All authors have read and agreed to the published version of the manuscript.

**Funding:** This research was funded by the Quzhou Science and Technology Plan Project, grant number 2022K108, and the Heilongjiang Provincial Natural Science Foundation of China, grant number YQ2022F014; and Basic Scientific Research Foundation Project of Provincial Colleges and Universities in Hei longjiang Province, grant number 2022KYYWF-FC05.

**Data Availability Statement:** As the algorithm in this paper relates to the project currently in progress, the project is private for some reasons, so the code is not convenient to be made public. If any reader wants to obtain the code of this article, you can send us an email. After receiving the email, we will determine whether to give the code to the reader after evaluation.

**Acknowledgments:** The authors acknowledge the Quzhou Science and Technology Plan Project (grant number 2022K108) and the Heilongjiang Provincial Natural Science Foundation of China (grant number YQ2022F014).

**Conflicts of Interest:** The authors declare no conflict of interest.

## Abbreviations

The following abbreviations are used in this manuscript:

| Short Name | Full Name |
| --- | --- |
| CNNs | Convolutional Neural Networks |
| AWGN | Additive White Gaussian Noise |
| DnCNN | Feed-forward Denoising Convolutional Neural Networks |
| NLF | Noise Level Function |
| BDE | Bayesian Deep Ensemble |
| DRNE | Deep Residual Noise Estimator |
| KPN | Kernel Prediction Network |
| MKPN | Multi-Kernel Prediction Networks |
| RED | Regularization by Denoising |
| WCNN | Wide CNN |
| MSR-block | Multiscale Residual Block |
| CBDNet | Convolutional Blind Denoising Network |
| NERNet | Noise Estimation module and Removal Network |
| SDNet | Software Defined Network |
| SRNet | Sparse Representation-based Network |
| MUS-ADD | Multi Scaling Aided Double Decker |
| ADMM | Alternating Direction Method of Multipliers |
| MS-WGSC | Multi-Scale Weighted Group Sparse Coding |
| AWTD | Adaptive Weighted Tensor |
| CRCNN | Cascaded and Recursive Convolutional Neural Network |
| KCRD | Kronecker Component with Low-Rank Dictionary |
| GRDN | Grouped Residual Dense Network |
| MSE | Mean Squared Error |
| PSNR | Peak Signal-to-Noise Ratio |
| SSIM | Structural Similarity |

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
