# Peer review of "Deep Signal-Dependent Denoising Noise Algorithm"

_electronics, doi:10.3390/electronics12051201_

Round 1
Reviewer 1 Report
This work presents a method for image denoising. The paper is well written, but its structure should be revised.
- The title should be revised. It did not represents the objective of paper.
- The references should following the crescent order.
- Insert the meaning of CNN, BM3D, KPN, SDNet, and other acronyms that are not explained on the text.
- There are references on arxiv, on other words, which is pre-print documents. It needs to change for peer review papers.
- Explain which is E operator on equations (7), (8), (9), and (10).
- Detailed the used deep neural network. Why a transfer learning approach was not applied?
- Table 1 and 2 are out of the template.
- The Figure 3 is confused. Please, revise them.
- Improve the quality of figures 6 and 7.
- A feasible comparison of this work and the related works should be performed. Authors only comment the results on “Experiment” section. Please, provide a section of methodology, results, and discussion.
- Authors should provide future works.
Reviewer 2 Report
In the manuscript, the authors proposed a two‐step denoising method, combining signal‐dependent noise Gaussian modeling and convolutional neural network. Two unavoidable errors in modeling, one generated in the estimation of noise parameters, and the other in calculation of pixel values via the local statistical feature, are reduced through the neural network.
The authors provided detailed descriptions of the signal‐dependent noise Gaussian modeling, which is appreciated, but missed some important information in convolutional neural network part. Certain terms in the manuscript are not clear or consistent and may reveal the risk of data leakage.
In the section of experiment, overall the results are well organized and convincing, but can be further improved if more details focusing on machine learning can be supplemented. Another flaw in this section is the lack of performance analysis on signal‐dependent noise Gaussian modeling.
In addition, there is room for improvement in language and grammar.
I believe a major revision is required before its quality can meet the requirements for publication in the journal.
Questions and suggestions:
1. Line 233. “The network takes the preliminary image and noisy image as input, and then extracts the shallow features of the image by a convolutional layer, and then extracts the deep features of the image through the residual network.” Figure 2 is not consistent with the description. There is no noisy image input shown in figure 2.
2. Line 285. “The noise level is fed into the network as an additional channel of input.” Again, there is no noise level input shown in figure 2. Also, the authors should make it clear whether the noise level input is an estimated one, or the exact noise level used to generate the noisy images. If the latter, it is a huge data leakage.
3. There is no training history shown at all. It can be added to Figure 5 as both take epoch as X axis.
4. The introduction of signal‐dependent noise Gaussian modeling is the main creation of the study. However, the performance analysis of this part is missing. How is the predicted noise level compared with the one used to generate noisy images?
5. Figure 6 and 7 are kind of meaningless and can be removed. Such kind of questionnaire could be misleading if the number of people attending is limited.
6. Abstract. “Extensive experiments demonstrate that our two‐step denoising method outperforms the state‐of‐the‐art methods for image denoising.” Although today many studies claim they outperform the SOTA, I hope all researchers can be humble and precise. Since the study is totally based on synthesized noisy images, at least more experiments based on wide varieties of unprocessed noisy images directly from camera should be tested before you claim SOTA.
Reviewer 3 Report
Contributions:
This paper proposes a two‐step denoising method. My comments are given below:
- The title of this paper is general. It should be renamed.
- (Page 5) Why eq. (9) can be transformed into eqs. (10)? How is the transformation error?
- (Page 6)Please create a layer graph and provide the model parameters for the CNN.
- (Page 6)The noise variance is small. I think the proposed method cannot work with heavy noise corruption. If so, the contribution of this method is limited.
- (Lines 312 to 314 on page 7)The statement is not correct. The CBDNet [12] outperforms the proposed method. In addition, the authors should discuss why the proposed method cannot outperform the CBDNet. The experiments for Set12 also have the same problem.
- The novelty of this paper is low, while the performance of the proposed method is comparable to the compared methods. Accordingly, the contributions of this study are not satisfied.
- (Page 6) Please define the loss function in sub-section 3.2.2.
- (Page 6) The CNN output should be the denoised image rather than the clean image in Fig. 2.
- (Page 7) Please create a table to present the training parameter.
- (Page 8)The local image for the original one should be provided in Fig. 3.
- (Lines 338 to 348 on page 9)The statement is not convincing. The denoised images in Fig. 8 are comparable. Please mark the differences.
- The keyword “denosing method” is not appropriate.
- (Line 62 on page 2)The noisy model should be expressed as an equation. In addition, x and y should be defined.
- (Line 69 on page 2) A reference was missed.
- (Line 77 on page 2)The symbol y denotes a noisy pixel. However, y represents a noise-free pixel in eq.(1). The symbol usage is inconsistent. The symbol x also has the same problem. Please check the symbol used throughout this paper.
- (Page 2)The meaning of the u and w should be defined in eq. (3).
- (Line 81 on page 2)” Xinhao Liu [5] et al. proposed…” should be revised as “Liu et al. [5] proposed…”. Please check the citation usage throughout this paper.
- This paper uses many abbreviations. Please create an abbreviation table in the appendix. It will help a reader read this paper.
- (Page 4) The operator D should be defined in eq. (6).
- (Line 215 on page 5)The word ”where” is missed at the beginning.
- (Line 225 on page 5)The second sentence is redundant. Please remove it.
- (Pages 9 and 10)The sub-grid lines should be removed in Figs. 5 and 6.
Reviewer 4 Report
In order to solve the problems in image denoising, such as poor noise suppression effect, smooth details and not flexible denoising ability, a two-step denoising method is proposed in this paper. After experimental verification, this method is superior to the most advanced image denoising methods at present, but there are also some problems in this paper. There are some suggestions for revision.
1. The motivation is not clear. Please specify the importance of the proposed solution.
2. Please highlight the contributions of the proposed solution in introduction.
3. There are some questions about the meanings represented by some formula letters in the article. For example, what does in formula 6 and formula 8 mean? For the convenience of readers, please check and explain the formula part.
4. Most of references are a little bit out of date. Please discuss more recently published solutions, such as "A Novel Multi-Modality Image Simultaneous Denoising and Fusion Method Based on Sparse Representation", Computers 10 (10), 129, 2021.
5. In this paper, an image restoration algorithm based on local statistical features is introduced. In this paper, the author does not explain what effect the algorithm has on the target task. The authors should give a supplementary explanation.
6. In the paper, the author mentioned the least square method to solve the two parameters and , but did not mention the specific solving process. Could the authors give a brief explanation on the specific process of parameter solving?
7. For convolutional neural network Figure 2, it is not clear why the authors adopt six same modules to extract features? Please explain the advantages of this design.
8. Please discuss how to obtain the suitable parameter values used in the proposed solution.
9. In the experimental part of the paper, it seems that the authors did not do ablation experiments to verify the effectiveness of each module in the network. Please add the ablation experiments and their results that the author thinks need to be done.
10. For the model designed in this paper, has the author considered doing experiments on more noisy and complex data sets, and whether the same experimental effect can be achieved as in this paper?
11. The experimental results are not convincing. Please compare the proposed solution with more recently published solutions.
Round 2
Reviewer 1 Report
The authors have addressed the resquested reviews and I found the paper suitable for publication. Please, follow the journal's template for acronyms.
Author Response
The acronyms have been revised.
Thank you!
Reviewer 2 Report
The authors manage to address all the comments/suggestions.
Author Response
Thank you for your recognition!
Reviewer 3 Report
Please mark and address the change the authors made. In addition, the response is not well prepared.
1. In my previous comment 4: The noise variance is small. I think the proposed method cannot work with heavy noise corruption. If so, the contribution of this method is limited. The authors only provide some statements. It is not sufficient. The authors should provide an example to show that the proposed method can work on large noise variance.
- In my previous comment 6: The novelty of this paper is low, while the performance of the proposed method is comparable to the compared methods. Accordingly, the contributions of this study are not satisfied. The authors only state the proposed method again. The novelty is not addressed.
3. In my previous comment 11: The statement cannot be convincing. The denoised images in Fig. 8 (revised version in Fig. 10) are comparable. Please mark the differences. However, the authors did not mark the difference.
Author Response
Thank you for your recognition!Here are my modifications.
1.In the paper, I added an extra set of experiments with high noise images.
2.The novelty of the paper has been added to the paper.
3.The difference in denoising is shown in Figure 11.
Reviewer 4 Report
All my concerns have been addressed. I recommend this paper for publication.
Author Response
Thank you for your recognition!
Round 3
Reviewer 3 Report
The quality of this paper has been improved by the authors. I think the paper can be accepted for publication.
Minor comment:
The sub-captions (a) and (b) should be revised as (a)-(d).
Author Response
I have revised the comment you put forward.
Thank you for your recognition!